# Enhancing sub-bandgap external quantum efficiency by photomultiplication for narrowband organic near-infrared photodetectors

Jonas Kublitski [1✉], Axel Fischer[1], Shen Xing[1], Lukasz Baisinger [1], Eva Bittrich [2], Donato Spoltore[1], Johannes Benduhn [1], Koen Vandewal [3] & Karl Leo[1,4✉]

Detection of electromagnetic signals for applications such as health, product quality monitoring or astronomy requires highly responsive and wavelength selective devices. Photomultiplication-type organic photodetectors have been shown to achieve high quantum efficiencies mainly in the visible range. Much less research has been focused on realizing near-infrared narrowband devices. Here, we demonstrate fully vacuum-processed narrow- and broadband photomultiplication-type organic photodetectors. Devices are based on enhanced hole injection leading to a maximum external quantum efficiency of almost 2000% at −10 V for the broadband device. The photomultiplicative effect is also observed in the charge-transfer state absorption region. By making use of an optical cavity device architecture, we enhance the charge-transfer response and demonstrate a wavelength tunable narrowband photomultiplication-type organic photodetector with external quantum efficiencies superior to those of pin-devices. The presented concept can further improve the performance of photodetectors based on the absorption of charge-transfer states, which were so far limited by the low external quantum efficiency provided by these devices.

[1] Dresden Integrated Center for Applied Physics and Photonic Materials (IAPP) and Institute for Applied Physics, Technische Universität Dresden, Dresden, Germany. [2] Leibniz-Institut für Polymerforschung Dresden e.V., Dresden, Germany. [3] Instituut voor Materiaalonderzoek (IMO-IMOMEC), Hasselt University, Diepenbeek, Belgium. [4] Center for Advancing Electronics Dresden (cfaed), Technische Universität Dresden, Dresden, Germany. ✉email: jonas.kublitski@tu-dresden.de; karl.leo@tu-dresden.de

From simple automatic lights in the halls of our buildings to the cruise control of cars, photodetectors (PDs) are playing a major role in everyday life[1]. Often, fast detection of faint signals is required, which is currently provided by inorganic avalanche photodiodes[2]. As the automotive industry is moving towards self-driving cars[3], properties like lower cost, higher sensitivity, wavelength selectivity, and form-free devices are required[4,5]. PDs made from organic semiconductors can offer these properties, but need further research to optimize these devices for low intensity signals, i.e., to reach high specific detectivities[6].

Photomultiplication-type organic photodetectors (PM-OPDs) are capable of amplifying small photocurrents without requiring external/additional circuit components. This can be achieved by a photo-induced enhanced injection via energy level bending caused by charge accumulation near the injecting electrode[7,8]. Following its observation in single material active layers[9] and donor-acceptor (D-A) heterojunctions[10], different strategies have been introduced to achieve such charge accumulation and thereby the required energy level bending through a lack of percolation path for one charge carrier type[11–14], energetic barriers via interfacial layers[15,16] as well as intentionally inserted trap states[17,18]. All of these strategies include an accumulation of one carrier type near the contact, such that the electrical field caused by these charges bends the energy level, enabling the opposite charge to be injected via tunneling across the injection barrier[7]. If the transit time of injected charge carrier is lower than the life-time of the accumulated, photo-generated charges, an EQE > 100% is observed. Here, we would like to stress that prior to the photomultiplication process the photon needs to be absorbed by the active layer. We therefore conclude that the minimum criteria for photomultiplication is that the internal quantum efficiency (IQE) is larger than unity.

The effect described above has been applied in organic and hybrid PDs, leading to external quantum efficiencies (EQEs) as high as $10^5\%$[19–21]. Nonetheless, the specific detectivity ($D^*$) achieved by these devices, which takes into account not only EQE but also the device noise current, ranges from $10^{10}$ to $10^{15}$ cm Hz$^{1/2}$ W$^{-1}$ (Jones) in the visible range[7], values comparable to those of diode-like OPDs. Guo et al. presented two different polymers blended with zinc-oxide nanoparticles, reaching $D^*$ of $10^{15}$ Jones in the ultraviolet region[17]. In the near-infrared (NIR) at 1200 nm, using colloidal lead sulfide (PbS) quantum dots, Lee et al. achieved $D^*$ of $10^{13}$ Jones[22], while $10^{14}$ Jones was attained for polymer-based devices in the visible range[23]. Recently, imagers[24] and dual band[25] OPDs were also fabricated using photomultiplication, with $D^*$ of ~$10^{14}$ and ~$10^{13}$ Jones, respectively. Moreover, photomultiplication has been also explored in perovskites[26], for which EQE of 4500% and $D^*$ of $10^{13}$ Jones were demonstrated at around 600 nm[27]. Despite the remarkable performance achieved by PM-OPDs in terms of increased EQE, limitations are still present in these class of devices. PM-OPDs suffer from high noise, a result of field dependent dark currents observed in these devices. In fact, this represents the main limitation in PM-OPDs as the gain acquired by biasing the device might also result in an increased dark current.

Photomultiplication has been extensively exploited in solution-processed organic/hybrid devices. However, despite the many advantages offered by sublimable small molecules, fewer examples were demonstrated in fully vacuum-processed devices[10,15,28–30]. Huang et al. demonstrated EQE higher than 1000% in devices based on $C_{60}$. These values were attributed to the disordered structure of $C_{60}$ and to interfacial traps at the interface $C_{60}$/hole transporting layer[31]. Similar results were achieved by interfacial blocking layers in hybrid (solution- and vacuum-processed)[18,32] and fully vacuum-processed devices[16,33], which are used to avoid

charge extraction, thereby causing the necessary band bending. In general, the vacuum deposition provides the possibility of depositing a vertical gradient of donor or acceptor molecules in the blend, as well as fine tuning the mixing ratio. Yet, such fine tuning extensively used in solution-processed PM-OPDs has not been investigated in vacuum-processed devices. Besides that, vacuum deposition offers the possibility of sequentially stacking of multiple layers, the well-established doping technology[34,35], straightforward fabrication of matrices of individual pixels, and is for commercial organic optoelectronic devices the currently preferred manufacturing technique.

Another aspect not considered in PM-OPDs concerns photo-multiplication in the extended charge-transfer (CT) state absorption region. With the aid of a Fabry-Perot microcavity, this rather weak absorption region has been used in NIR narrowband organic photodetectors (CT-OPDs)[36–38]. Such narrowband OPDs could significantly benefit from the increased IQE, if photo-multiplication would take place also by direct excitation of CT states. However, it is unclear whether direct excitation of CT states can result in a photomultiplication process. Utilizing the intermolecular CT state absorption renders possible to detect NIR photons beyond 1700 nm meanwhile using rather small and sublimable organic semiconductors where the absorption profile can be easily tuned by the D-A system. On the other hand, the weak intermolecular absorption cross section[39,40] challenges the overall performance and here the photomultiplication could improve the electrical performance by increasing the gain for every absorbed photon.

Here, we report a fully vacuum-processed PM-OPD based on low-acceptor content (3 wt%) ZnPc:$C_{60}$ material system, with a maximum EQE of almost 2000% achieved at −10 V. Additionally, an optimum operation regime maximizing EQE while keeping a low dark current is found, leading to a $D^*$ of $2.2 \times 10^{12}$ Jones at 670 nm. Sensitively measured EQE spectra reveal that direct excitation of CT states also results in photomultiplication, which is confirmed by an IQE higher than 100% over the entire absorption region. Under −5 V reverse bias, the EQE of the PM-OPD surpasses that of an optimized pin-photodiode, demonstrating the potential for application in microcavity CT-OPDs. Indeed, by exchanging the transparent ITO contact with semi-transparent Ag mirrors, while varying the thicknesses of the optical microcavity from 355 to 400 nm, peaks in the NIR region originating from cavity enhanced CT absorption arise. Narrow-band PM-OPDs show peak EQEs from 20 to 80% under −10 V with full width at half maximum (FWHM) from 20 to 40 nm, and $D^*$ of around $10^{11}$ Jones for all the resonant wavelengths. These results are comparable with narrowband organic pin-photodiodes based on cavities[36,38], and higher than that of narrowband photomultiplication-type devices based on charge injection narrowing (CIN)[41]. The concept presented here can be used to boost EQE of CT-OPDs, which so far was mainly limited by the low absorption cross section of CT states, the low internal quantum efficiency[38], and the parasitic absorption of the contacts.

## Results

Controlling the mixing ratio is essential for the working principle of previously reported PM-OPDs. For enhanced hole injection in reverse bias, electrons must accumulate near the cathode: we designed our devices based on a low-acceptor content (3 wt%), such that few percolation paths are formed. The well-known ZnPc:$C_{60}$ system is chosen given the LUMO energy offset between these materials. At this concentration, electrons are intentionally trapped within the LUMO level of $C_{60}$ and the bending caused by electron accumulation in the $C_{60}$ phase leads to EQEs above 100%. Below, we describe how this can be

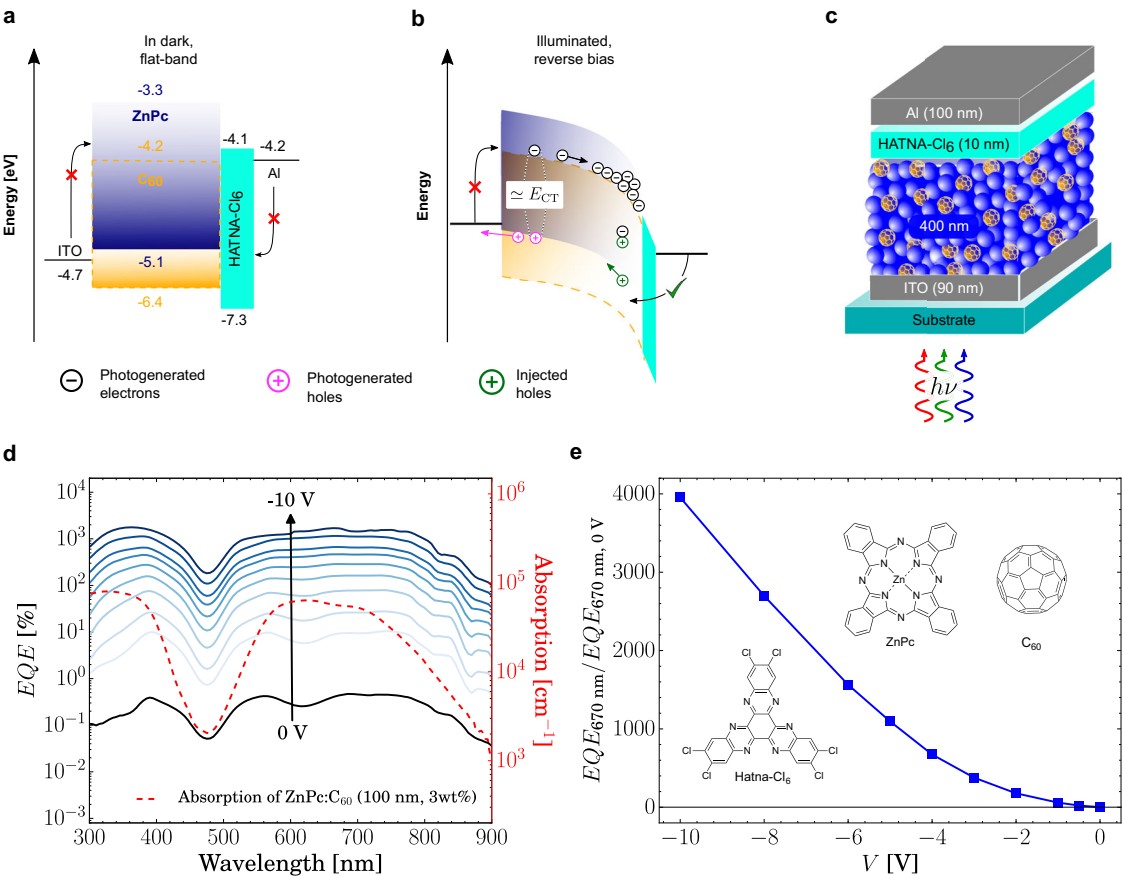

**Fig. 1 Operation, device structure, and EQE of a PM-OPD.** Schematic energy diagram **a** under dark at flat band condition and **b** negatively biased under illumination. **c** schematic device structure. **d** Voltage-dependent EQE (solid lines) of the device shown in **c** comprising ZnPc blended with $C_{60}$ at 3 wt%. Each line corresponds to one symbol in **d**. Dashed red line shows the absorption spectrum of the same blend. Additionally, in **e**, the relative enhancement factor as a function of applied reverse bias is presented. Symbols show the ratio between EQE at 670 nm at each voltage from **d** normalized by the EQE at 670 nm at 0 V. The blue line is a guide to the eye. Note that no saturation is observed, indicating that EQE can be further increased. Inset shows the chemical structure of $C_{60}$, ZnPc and HATNA-$Cl_6$. The energy level values of ZnPc, $C_{60}$, and HATNA-$Cl_6$ in **a** are taken from the literature[55–57].

achieved in this system and how this effect can be used in narrowband PM-OPD.

**Enhancing the external quantum efficiency.** The PM-OPD operation in dark and under light as well as the architecture are shown in Fig. 1a–c. The bulk heterojunction comprising low $C_{60}$ content (3 wt%) is sandwiched between two contacts, top Aluminum and bottom ITO, from which light enters the device, as shown in Fig. 1c. Pristine HATNA-$Cl_6$ is used as hole blocking layer (HBL) in between active layer and Al electrodes to reduce the reverse dark current. However, the thickness of HATNA-$Cl_6$ must be carefully controlled such that injection is enabled upon band bending. Under reverse bias in the dark, the high injection barriers (Fig. 1a) hinder holes and electrons to be injected into ZnPc and $C_{60}$, respectively. Under forward bias, electrons and hole are injected into the device, such that the device behaves similarly to a photodiode. When the device is exposed to illumination, excitons are formed, and free charges are generated at D-A interface. Due to absence of percolation paths, electrons are trapped in the acceptor phase and their transport is further hindered by the low electron mobility of the electrically undoped HATNA-$Cl_6$ layer[42]. While n-doped HATNA-$Cl_6$ has been already employed as an electron transport layer, in this device, we intentionally use a pristine layer such that the electron extraction is hindered and slowed down, which helps the electron accumulation at the cathode. This accumulation of electrons upon

illumination causes that the energy levels bend in the vicinity of the contact, enabling holes from the external circuit to tunnel through the energy barrier imposed by the HATNA-$Cl_6$ layer into the donor phase, where they are efficiently transported together with the photo-induced holes towards the anode.

From the process described above, a voltage-dependent increase in EQE is expected, as higher reverse voltage further decreases the energy barrier for injection. The black line in Fig. 1d shows the EQE measured at 0 V, for which a maximum of 0.5% is achieved. This rather low value can be explained by the interrupted percolation path for electrons[43] and charge separation probability at this concentration as well as by the unoptimized device architecture for 0 V operation. Clearly, no photomultiplication is observed. To prove the described working mechanism, we increase the reverse applied bias from 0 to −10 V. Indeed, the EQE rises accordingly, reaching almost 2000% at −10 V. Further increase is expected for higher negative bias, as can be seen from Fig. 1e, where no saturation in the relative enhancement is observed. However, as it will be discussed below, an optimum operation regime exists in the range of −2.5 V, where the highest $D^*$ is achieved.

In spite of the extensive work performed by different groups on PM device structures[10,16], this effect has not yet been utilized to increase EQE in the spectral region of CT state absorption. In D-A systems, interaction between donor and acceptor results in an extended but weak absorption band related to an optical transition from the HOMO of the donor to the LUMO of the

acceptor. Recently, enhanced CT state absorption photodetectors (CT-OPDs) have been introduced[36–38], which could benefit from high gain for absorbed photons provided by photomultiplication. However, it is not clear whether photomultiplicative gain can be achieved for photons that directly excite the CT states. Before investigating PM gain in the CT absorption band, we first determine the optimum D-A concentration to achieve PM, as well as the relation with dark current. Below, we investigate these issues in ZnPc:$C_{60}$ based devices.

**Effect of acceptor concentration on the photomultiplication.** In the previous section, we showed that EQE can be enhanced by three orders of magnitude by photomultiplication based on electron accumulation. In polymeric systems based on the same effect, it is well accepted that low concentration of one material type (D or A) is necessary for attaining photomultiplication, a condition which has not been investigated for small molecule based devices. Zhang et al. reported an efficient photovoltaic effect at around 5 wt% donor content, which suggests that at such concentrations, hole transport takes place efficiently[44]. The minimum acceptor concentration required for a BHJ to work as a D-A photodiode has not been established for low-acceptor content systems.

To investigate the concentration dependence, we fabricated devices comprising concentrations from 1 to 4 wt%. The results are depicted in Fig. 2. Devices comprising 1 wt% and 2 wt% mixing ratios do not show any amplification and behave as an unoptimized photodiode. For these devices, EQE does not overcome 100% and is limited by the poor free charge carrier generation of the system, which explains the slightly higher EQE of the blend at 2 wt%, where more exciton dissociation centers are available.

At 3 wt%, the photocurrent increases almost one order of magnitude at a given reverse voltage. This abrupt enhancement is a result of a sufficient accumulation of charges at the contact, leading to an increased injection. At 4 wt%, the device still shows amplification, but the performance of the device deteriorated, which we attribute to a more efficient extraction of electrons at this concentration. If the concentration is further increased, percolation paths are formed and efficient extraction of both charge carrier types takes place. The device will then behave as a typical organic photodiode. From these results, we infer that an optimum concentration exists (in our case, 3 wt%), where sufficient charges are trapped to cause energy level bending while providing enough free charge carrier generation. At concentrations higher than that, the injection is reduced either because

charges are extracted more efficiently or because the bimolecular recombination rate increases. The maximum amplification found at a very specific concentration demonstrates the importance of highly controlled mixing ratios and morphology achievable in vacuum-processed devices.

The dark current of the 1 wt% device is lower than that of the 2 wt% device, which we attribute to the smaller amount of D-A interfaces as well as to an increased number of traps[45–47]. However, comparing the dark current of 3 wt% and 4 wt% devices, we see that the former has a higher dark current and therefore a different behavior than the devices comprising 1 wt% and 2 wt%. Analyzing the four devices together, we observe that an enhanced photocurrent, and thereby EQE, seems to be correlated with an increased dark current. Daanoune et al. suggested that this correlation is an intrinsic consequence of the working principle of devices based on enhanced injection by charge accumulation[48]. In the dark, charges are thermally activated over the bandgap of the system, which in a D-A heterojunction corresponds to energy of CT states. In an ideal diode, this current corresponds to the saturation current, $J_0$[46]. In PM-OPDs, charge carriers forming $J_0$ accumulate in the same way that photo-generated carries do, leading to enhanced injection also of dark current. In the same study, the authors also correlate the rather slow speed of PM-OPDs to the slow trap dynamics. This not only explains the observed trend, but also indicates that this dark current effect might be detrimental for the specific detectivity ($D^*$) of PM-OPD. This aspect is addressed in the following section.

**The role of dark current in Photomultipliers.** The specific detectivity $D^*$ of photodetectors depends on the EQE, which, as shown, can be enhanced by photomultiplication. However, it also depends on the noise of the device. We can express $D^*$ in terms of the device spectral noise density, $S_n$, as follows:

$$D^* = \frac{q\lambda}{hc}\frac{EQE}{S_n}.$$

(1)

where $q$ is the elementary charge, $\lambda$ the wavelength, $h$ the Planck constant, and $c$ the speed of light. In the absence of a frequency dependent component of the noise, $S_n$ reads:

$$S_n = \sqrt{2qJ_D + \frac{4k_BT}{R_{sh}}},$$

(2)

which takes into account the shot and thermal noise, the first and second term in Eq. 2, respectively, where $J_D$ is the dark current density, $k_B$ the Boltzmann constant, $T$ the temperature and

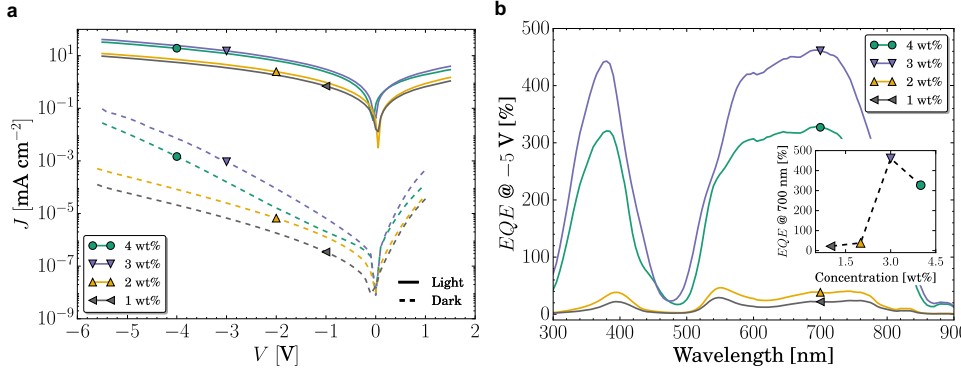

**Fig. 2 Electrical characteristics of PM-OPD. a** JV curves in dark (dashed lines) and under 100 mW cm$^{-2}$ illumination (solid lines) of devices based on ZnPc:$C_{60}$ at different concentrations. A clear increase of the light and dark current is observed for devices with concentration higher than 3 wt%. **b** EQE of the same devices showed in **a**, measured at −5 V. The observed increase in current under illumination results in an increased EQE over 100%, achieving maximum of 450%.

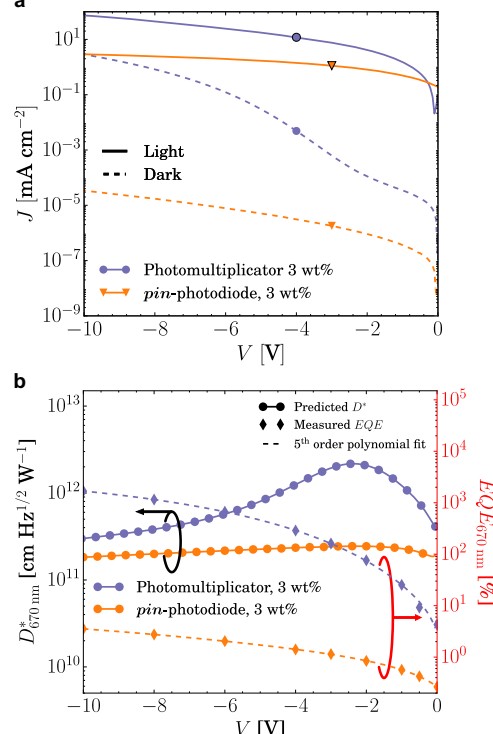

**Fig. 3 Comparison of a PM-OPD and an equivalent pin-photodiode. a** *JV* characteristics in dark and under 100 mW cm$^{-2}$ illumination of a pin-photodiode and of a PM-OPD. **b** The EQE measured at different voltages is fitted with polynomial function from which the detectivity is predicted. The total noise is calculated from the shot and thermal noise, which are obtained from the dark current and the shunt resistance at room temperature, respectively. The PM-OPD shows a maximum $D^*$ of $2.2 \times 10^{12}$ Jones around $-2.5$ V, which is higher than $D^*$ of the equivalent pin-photodiode over the entire range measured, besides the high dark current at high reverse bias. The pin-photodiode comprises the same structure shown in Fig. 1c, except for the ETL, where instead of HATNA-Cl$_6$, BPhen (8 nm) is used, see Supplementary Fig. 1b.

$R_{sh}$ the shunt resistance normalized by the area (m$^2$ Ω) extracted from the inverse of the derivative of *JV* curve around 0 V. In most organic devices, however, the high reverse dark current makes the shot component the main source of noise, which represents a limitation also in diode-like organic photodetectors.

In polymer-based devices, different material systems have been reported to show high EQE; however, the values of the dark current have not always been presented. As mentioned before, the increase in photocurrent is usually correlated with an increased dark current. Therefore, both parameters have to be analyzed concomitantly in order to identify whether photomultiplication can be used to indeed get an increased $D^*$ as an equivalent pin-photodiode. In Fig. 3a, the dark current of the same devices shown in Fig. 2a is compared to that of a pin-photodiode comprising the same concentration. As can be seen, while the photocurrent reaches values two orders of magnitude higher than that of the pin-photodiode at $-10$ V, the dark current is four orders of magnitude higher, see Fig. 3a. Therefore, in order to overcome the performance of a pin-photodiode in terms of signal detectivity, EQE has to be as high as possible to even compensate such an increase in dark current. We have already shown that EQEs of almost 2000% can be achieved for small molecule devices. Now we must investigate if $D^*$ is indeed higher than that of equivalent pin-photodiodes.

In most well working pin-photodiodes, EQE is weakly dependent on applied bias, which can also be seen in Fig. 3a, where the photocurrent does not increase significantly with increasing reverse bias. Nonetheless, we measured the voltage-dependent EQE (in Fig. 3b) and approximated its maximum value as function of voltage by a 5th order polynomial, which allows to have estimated values of EQE for every voltage. From the fit, together with the measured dark current and shunt resistance, the voltage-dependent $D^*$ can be calculated, according to Eq. (1) and Eq. (2). The same procedure is used for the PM-OPD. The results are compared in Fig. 3b.

From Fig. 3 it is obvious that increasing EQE only is not sufficient to achieve high detectivities. As the dark current usually changes by orders of magnitude as a function of applied bias, the latter dominates $D^*$. Given this tradeoff, an optimum operation region has to be found, where the effect of the dark current does not overcome the enhancement in EQE. In the most favorable operation region, $D^*$ of $2.2 \times 10^{12}$ Jones is obtained for the photomultiplier device, comparable to results reported for PM-OPDs and higher than $D^*$ provided by the equivalent pin-photodiode, where $D^*$ is in the order of $10^{11}$ Jones over the entire range measured. Whether the abrupt increase in dark current is indeed an intrinsic consequence of the photomultiplication process, as previously suggested, is still not clear. While it seems to be the case for most reported photomultipliers, some examples combine a high EQE with low dark currents, leading to high performance[17]. If the dark current of the PM-OPDs would be comparable to the one obtained in the pin-photodiode, $D^*$ could be improved by two orders of magnitude. This shows that further investigations are needed to understand the origin of dark current in this device class.

**Enhancement of charge-transfer state absorption/response in photomultipliers**. We have shown that by controlling the D-A mixing ratio, photomultiplication can also be achieved in vacuum-processed organic blends in the spectral range of strong donor absorption. Whether the same effect is present when exciting in the CT absorption region is an important and, so far unaddressed question, which is relevant for microcavity CT-OPDs. However, the low-acceptor concentration required for photomultiplication to take place decreases the number of D-A interfaces, establishing a tradeoff between enhanced EQE and absorption. In order to be useful, the amplified EQE in the CT absorption region should overcome the EQE of a standard photodiode-based device, in which a higher density of CT states is present.

In Fig. 3, we showed that the photomultiplier is superior the pin-photodiode based on the same blend ratio. However, ZnPc:C$_{60}$ pin-photodiode was shown to produce maximum photo-current when used at 50 wt% mixing ratio[49]. Therefore, in Fig. 4, the sensitively measured EQE spectra of a PM-OPD (3 wt%) at different bias are compared to that of a standard ZnPc:C$_{60}$ (50 wt%) pin-photodiode at zero bias. The CT band is observed for wavelengths longer than 800 nm in the EQE spectra of both devices. In the pin-photodiode, the CT band is more pronounced due to the higher density of CT states provided by the larger amount of D-A interfaces. In the same region, the PM-OPD shows a lower absorption shoulder, but also extending to the near-infrared region. As expected, at zero bias, the EQE of the PM-OPD is orders of magnitude lower than that of the pin-photodiode, as no enhanced injection takes place. By applying $-2$ V, however, the EQE in the visible range overcomes that of the pin-photodiode demonstrating the enhanced injection upon illumination. When -5 V are applied to the device, the entire EQE spectrum of the PM-OPD surpasses that of the pin-photodiode,

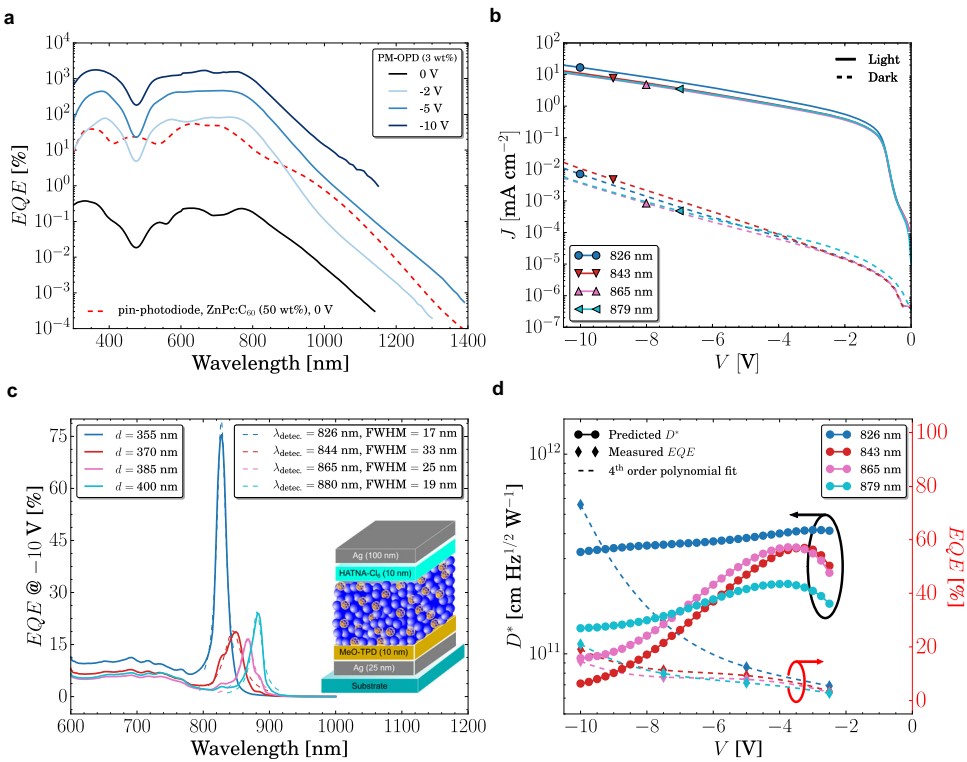

**Fig. 4 Photomultiplication in the CT region and used in narrowband devices. a** EQE as a function of excitation wavelength for the broadband photodetector. Red dashed line shows the EQE spectrum of a conventional ZnPc:$C_{60}$ (50 wt%) photodiode and the black solid line the PM-OPD, both at 0 V. Solid lines show the spectra of the PM-OPD (3 wt% $C_{60}$) under bias as indicated in the legend. Under −5 V, the higher injection provided by the energy level bending leads to an EQE higher that of a conventional pin-photodiode. **b** JV characteristics under dark and under 100 mW cm$^{-2}$ of four different narrowband devices with varying resonant wavelength as indicated in the legend. **c** EQE of cavities of the same devices shown in **b** at −10 V. As the active layer thickness increases from 355 to 400 nm, the resonant wavelength redshifts from around 830 to 880 nm. Dashed lines show the fit to a Lorentzian function, from which the FWHM is extracted. **d** EQE measured at different voltages is fitted with polynomial function, from which $D^*$ is predicted. An optimum operation region is found around −3.5 V, where $D^*$ of $6 \times 10^{11}$ Jones is obtained.

confirming that direct excitation of CT states can also trigger the photomultiplication process in these devices.

While the PM effect is commonly accompanied by an EQE above 100%, it is the IQE which better defines the physical phenomenon behind this effect. In order to induce PM, free charge carriers must be firstly generated, requiring photons to be absorbed. As a means of quantifying whether absorbed photon induce enhanced injection in the sub-gap absorption region in our devices, we estimated their IQE, which accounts only for absorbed photons. To that end, we employ the transfer matrix method[50,51] (TMM) using ellipsometrically derived $n,k$-values to simulate the absorption in our devices and adjust the IQE to reproduce the magnitude of the measured EQE spectra, see Supplementary Fig. 6 for more details. Under −10 V a constant IQE of 1750% over the full wavelength range, i.e., including CT absorption, is required to describe the experimental data as shown in Supplementary Fig. 1. Figure 4a together with TMM simulation data demonstrate the potential of combining such systems with optical cavities to accomplish high performance narrowband photodetectors.

In order to test whether such devices could be achieved, we embedded the best performing PM-OPD, i.e., 3 wt%, into an optical microcavity, see inset in Fig. 4c for the device structure. Due to the higher Ag work function as compared to that of ITO, we inserted a 10 nm-thick MeO-TPD layer to hinder hole injection in reverse bias. With aid of TMM, we simulate the optical photoresponse of a device comprising the same active layer thickness of 400 nm, which leads to a resonant peak around

880 nm. Different resonant peaks can be achieved by varying the thickness of the active layer, leading to tunable near-infrared detection as shown in Supplementary Fig. 3. The JV and EQE characteristics of devices comprising thicknesses from 355 to 400 nm are shown in Fig. 4b, c, respectively. As predicted by the optical simulation, narrowband peaks arise in the EQE spectra. As a demonstration, we tune the response wavelength from ~830 nm to ~880 nm, which under −10 V, reaches maximum EQE of 20–80%, with a FWHM varying from 20 to 40 nm. As to prove that photomultiplication also takes place in the narrowband devices, we estimate the IQE of these devices. Indeed, for the device with a detection wavelength of 828 nm, an IQE of 160% is achieved. The three other devices show IQE of around 40%, from which it is not possible to infer whether such values are a result of the PM effect or other phenomena. Therefore, to elucidate that, we increase the bias voltage to −15 V. This leads to EQEs and IQEs above 100% for all four devices, with peak values of ~430% and ~920%, respectively, see Supplementary Fig. 4.

Also in the microcavity devices, the dark current plays an important role in the final $D^*$. Although in these devices a better on/off ratio is kept along the reverse bias region as compared to those of Fig. 2, see Fig. 4b, the on/off ratio decreases as the reverse voltage increases, pointing to a decreased $D^*$ at high reverse bias. Therefore, we also estimate an optimum operation regime, where the tradeoff between EQE and dark current is maximized. As depicted in Fig. 4d, we obtain $D^*$ as high as $6 \times 10^{11}$ Jones in narrowband devices, which is comparable to narrowband pin-devices based on cavities[36,38]. Moreover, it is superior than that of

narrowband photomultiplication-type devices based on CIN[41], where, in addition, excessively thick devices are demanded, which increases the operation voltage.

**Transient photocurrent**. Another important figure-of-merit of photodetectors is the response speed. In PM-OPDs, the temporal response is believed to be limited by the trapping/detrapping dynamics[17,48], while other processes such as charge carrier transit time should be much shorter. In order to investigate the response speed of our devices, transient current measurements are performed. The rise time (from 10 to 90% of the device saturated signal) and fall time (from 90 to 10% of the device off signal) are summarized in Supplementary Table 1. The rise time of both broad- and narrowband devices ranges from 20 to 600 µs, corresponding to $-3$ dB cutoff frequencies of ~19.5 to ~0.4 kHz. These values are comparable to the best performing PM-OPDs reported so far[7] and are suitable for health monitoring and video-frame-rate imaging applications.

## Discussion

In CT-OPDs, the thicknesses required are much smaller than those used in narrowband devices based on charge collection narrowing (CCN)[52,53] or on CIN[41,54]. In the latter, for example, the thickness of the active layer must be much larger than the inverse of the absorption coefficient of the active layer, such that under illumination charges are generated close to the injecting contact, thereby causing the necessary band bending[54]. Moreover, in CT-OPDs the response can be further redshifted not only by increasing the thicknesses of the active layer, but also by introducing spacer/transport layers, which make the device electrically thin, but optically thick[36–38]. By combining the concept used in CT-OPDs with photomultiplication, we are able to achieve much thinner devices, as demonstrated in Fig. 4b–d, where active layers of 355 nm were used for spectral response at ~830 nm, compared to 2.5 µm reported for spectral response at 650 nm when using CIN combined with photomultiplication[41]. The concept presented here can further benefit from the properties of microcavity devices while keeping enhanced EQE by photomultiplication at reasonable thicknesses and operation voltages. Moreover, in PM-OPDs, the position of the active layer can be placed in an optimized position, either near the contact to enhance injection or such that optical overtones are minimized. There are systems combining low dark currents with enhanced EQE[17], which, together with our concept, can potentially overcome the performance of state-of-art near-infrared narrowband devices.

In summary, we investigate the photomultiplication effect in fully vacuum-processed organic photodetecting devices. At 3 wt% of $C_{60}$, a significant increase in EQE is observed under reverse bias, attributed to the accumulation of electrons caused by the lack of percolation paths. In the optimum operation regime, a specific detectivity $D^*$ of ~$10^{12}$ Jones is achieved. In addition, sensitively measured EQE spectra reveal that the enhancement extends to the CT absorption region, which indicates that these states also trigger photomultiplication, making microcavity CT-OPDs with photomultiplication possible. Indeed, by exchanging top and bottom contacts by semitransparent mirrors, narrowband NIR PM-OPDs with response from 830 to 880 nm can be realized, achieving $D^*$ of ~$10^{11}$ Jones and FWHM as low as 20 nm. The combination of optical microcavities with the photomultiplication effect can potentially boost NIR CT-OPDs, which so far were limited by the low EQE in the CT absorption region. Furthermore, much thinner devices are sufficient to achieve narrowband detection, as compared to the CIN approach. Additionally, the method presented here allows placing the active layer in different positions within the device or using gradients of D-A mixing ratio, thereby enhancing injection and diminishing the effect of optical overtones, a critical problem in CT-OPDs.

## Methods

**Device preparation**. Organic layers used in the devices were thermally evaporated on glass substrates covered by pre-structured ITO contact (32 Ω.$\square^{-1}$, Thin Film Devices) at ultrahigh vacuum (pressure < $10^{-7}$ mbar). Before deposition, substrates are cleaned for 15 min in different ultrasonic baths with NMP solvent, deionized water and ethanol followed by $O_2$ plasma for 10 min. Organic materials were purified 2–3 times via sublimation. The overlap of the bottom and top contact (Al, 100 nm, Kurt J. Lesker) defines the device active area (6.44 mm²). After evaporation, samples are directly transferred to a glovebox with inert atmosphere, where they are encapsulated with a cover glass, fixed by UV hardened epoxy glue. A moisture getter (Dynic Ltd.) is inserted between top contact and the glass to hinder degradation.

**Current-voltage characteristics**. Illuminated $JV$ characteristics were measured using a source measurement unit (Keithley SMU 2400). The devices were illuminated at intensity of 100 mW cm$^{-2}$ provided by a sun simulator (Solarlight Company Inc., USA). The intensity is calibrated by a Hamamatsu S1337 silicon photodiode. Dark $JV$ characteristics were measured with a sensitive SMU (Keithley SMU 2635). Every measurement data point was acquired after steady-state conditions were achieved. The measurement is controlled by the software SweepMe! (https://sweep-me.net/).

**External quantum efficiency (EQE)**. The current generated by the device under monochromatic light chopped at 170 Hz (Oriel Xe Arc-Lamp Apex Illuminator combined with Cornerstone 260 1/4 m monochromator (Newport, USA)) is measured with a lock-in amplifier (Signal Recovery SR 7265). A mask (2.78 mm²) is used to avoid edge effects. The same procedure is followed to monitor the light intensity, measured with a calibrated silicon photodiode (Hamamatsu S1337 calibrated by Fraunhofer ISE). EQE is obtained by the ratio of charge carriers generated by the device with the number of incoming photons.

**Sensitive external quantum efficiency (sEQE)**. A chopped monochromatic light (140 Hz, quartz halogen lamp (50 W) used with a Newport Cornerstone 260 1/4 m monochromator) is shined to the device. The current generated at short-circuit conditions or at applied bias is fed into a current-voltage preamplifier (DHPCA-100, FEMTO Messtechnik GmbH) before being measured by a lock-in amplifier (Signal Recovery 7280 DSP). The time constant of the lock-in amplifier was chosen to be 500 ms and the amplification of the preamplifier was increased to resolve low photocurrents. Light intensity was obtained by using a calibrated silicon (Si) and indium-gallium-arsenide (InGaAs) photodiode.

**Transient current measurements**. To record current transients, the measured device was held at short-circuit, connected to the low impedance (50 Ω) input of an oscilloscope (DPO7354C, Tektronix). 100 Hz square waveform from a signal generator (Agilent 33600 A Series) was used to control a MOSFET (IRF630N) driving the high-power white LED (LED Engin, Osram Sylvania Inc.). The pulse length was set to 5 ms, allowing device to reach a steady state before switching off the light. The signal from the device was pre-amplified (DHPCA-100, FEMTO Messtechnik GmbH) prior to being recorded on the oscilloscope.

**Ellipsometry**. Variable-angle spectroscopic ellipsometry was performed with an M2000 UI (J.A. Woollam Co., Inc., Lincoln, USA, wavelength range: 245–1690 nm). The uniaxial optical dispersion of a 100 nm ZnPc layer doped with 3 wt% $C_{60}$ was obtained using an optical model (Si/SiO$_2$(1 µm)/ZnPc:$C_{60}$(100 nm, 1 Tauc-Lorentz and 5 Gaussian oscillators, energy positions coupled $z$ to $xy$) with sharp interfaces and additional EMA (50% void, 50% ZnPc:$C_{60}$) roughness top-layer.

## Data availability

All data generated in this study have been deposited in the Materials Cloud database under accession code https://doi.org/10.24435/materialscloud:8w-q6.

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

## Acknowledgements

J.K. acknowledges the German Academic Exchange Service for the Ph.D. fellowship. J.B. acknowledges the DFG project VA 1035/5-1 (Photogen) and the Sächsische Aufbaubank through project no. 100325708 (InfraKart). E.B. thanks Roland Schulze (IPF) for performing the ellipsometry measurements. L.B. acknowledges the European Union's Horizon 2020 research and innovation program under Marie Skłodowska-Curie Grant Agreement number 722651 (SEPOMO).

## Author contributions

J.K., A.F., D.S., K.V., and K.L designed the experiments, prepared the detectors, and optimized the devices. J.K. and S.X. performed the standard characterization of detector and simulated the absorption spectra. L.B. measured the transient photocurrent and analyzed the data. E.B. analyzed the ellipsometry data and extracted the $n,k$-values used in the TMM simulations. J.K. and J.B. measured sensitive EQE spectra. A.F., D.S., and J.B., supervised their team members involved in the project, K.V. and K.L. supervised the overall project. All authors contributed to analysis and writing.

## Funding

## Competing interests

A.F. is co-founder of "Axel Fischer und Felix Kaschura GbR", which provided the measurement software "SweepMe!"(sweep-me.net). The name of the program is mentioned in the experimental of the manuscript. The other authors declare that they have no conflict of interest.
