## [Peer Review File · Nature Communications]

Reviewers' comments:

Reviewer #1 (Remarks to the Author):

Kublitski et al. reported demonstrate fully vacuum-processed narrowband and broadband PM-OPDs with ZnPc:C60 (100:3, wt/wt) as the active layers. The active layers have single carrier transport channels and some isolated electron traps formed by ZnPC/C60/ZnPc. The photogenerated electrons will be trapped in C60, which will induced the interfacial band bending for hole tunneling injection from external circuit. Devices are based on enhanced hole injection leading to an maximum external quantum efficiency (EQE) of almost 2000% at -10 V for the broadband device. I am pleased to recommend this manuscript potential publication in Nature Communication after well addressing the following issues, please note that the statement on PM in Near infrared region is wrong due to the EQE values are much less than 100%.

1. The authors claimed that "Narrowband PM-OPDs show peak EQEs from 20 to 80% under -10 V with full width at half maximum (FWHM) from 20 to 40 nm" is wrong. If the EQEs are lower than 100%, which is not a kind of PM-OPDs, should be photodiode type organic photodetectors.

2. The authors also claimed "narrowband photomultiplication type devices based on charge collection narrowing (CCN)³³" The reference [33] reported narrowband photomultiplication type organic photodetectors based on charge injection narrowing (CIN) concept, rather than CCN concept. The detailed description on CIN concept is given in Adv. Opt. Mater. 2018, 1800249. The difference between CCN and CIN should be carefully discussed for the reader better understand. The OPDs based on CCN also exhibit EQE much less than 100%, the narrowband is obtained at the price of EQE. The narrowband prosperity of OPDs based on CCN will lose under large bias. For the OPDs based on CIN, the EQE will be markedly increased along with the increase of bias, while keeping the FWHM as constant. The typical works on PM-OPDs, Org. Electron. 2019, 69, 354, Nanoscale 2019, 11, 16406 should be cited, the active layers also have single charge carrier transport and some isolated the opposite charge traps.

3. The authors should investigate the OPDs with ZnS/ZnS:C60 (100:3, wt/wt) as the active layers, I guess that the double layers OPDs will exhibit different spectral range or a narrowband response. The injected holes will be more efficiently transported in ZnS layer, leading to the enhanced EQE values.

4. For the NIR OPDs, the EQE values are lower than 100%, which should be photodiode type OPD, rather than PM-OPDs. The authors should carefully revise the corresponding statement. The recent review article Laser Photonics Rev. 2020, 2000262 should be cited about the development on broadband organic photodetectors, including photodiode type and photomultiplication type.

5. The authors should pay more attention on the reference [33] mentioned many times in your manuscript, the working mechanism is not CCN, should be CIN (Charge injection narrowing). The PM should be attributed to interfacial traps induced interfacial band bending for hole tunneling injection from external circuit.

6. The caption of Fig. 4 " Photomultiplication in the CT region" is wrong, it is apparent that the EQE values are much less than 100%. The authors should carefully revised the related points, especially for the title of this manuscript "... Photomultiplication in Narrowband Organic Near-Infrared Photodetectors".

Reviewer #2 (Remarks to the Author):

The authors presented a method to enhance the external quantum efficiency (EQE) of near-infrared (NIR) photoresponse. This is realized by utilizing the charge-transfer states between organic electron donor molecule (D, ZnPC) and electron acceptor molecule (A, C60) with the photodiode having a photomultiplication-type structure, that is, reducing the acceptor composition to a few percent (e.g., 3 wt% here). The work mainly divided into two parts. The first part is to demonstrate the photomultiplication effect can be achieved with the active layer having 3 wt% C60 in ZnPC. A maximum EQE of almost 2000% at -10V and the optimal specific detectivity D^* of 2.4×10^{12} Jones at about -2.5 V were demonstrated. The idea of on-purpose reducing acceptor composition in either polymer:fullerene or polymer:semiconductor nanoparticle active layer to achieve electron-trapping induced band bending and hole injection to realize photomultiplication has been reported before (e.g., refs. 10 and 14). Therefore, the first part of the work is an increment of the previously published work. The second part of the work is to utilize this photomultiplication-type photodiode (mainly the active layer with 3 wt% C60) but with semi-transparent Ag bottom electrode and opaque Ag top electrode to form nanocavity to realize the NIR photoresponse due to charge-transfer states. By varying the active layer thickness, the wavelength of NIR response varies so does EQE. The highest EQE was achieved $\sim 80\%$ at the peak position of 826 nm at -10V applied bias. Utilizing charge-transfer states of ZnPC:C60 and nanocavity to realize NIR detection was reported by the same group three years ago at Nature Communications (ref. 30). The difference between this work and the previous one is mainly the composition change (from roughly ZnPC:C60 of 1:1 ratio to 3 wt% C60) and active layer thickness change (from < 100 nm to 300-400nm). If compare the photoresponse at the same wavelength, e.g. ~ 870 nm, EQE is $\sim 20\%$ (ref. 30 Fig. 2a), and EQE is $\sim 20\%$ (this work, Fig. 4C). Therefore, the second part of the work is also an increment of the previous published work by the same group. More comments are provided below.

1. The PM-OPDs discussed in Introduction are mainly polymeric donors, few are small molecule donors (or not explicitly discussed). This work used small molecule as donor. So, it would be helpful a review of the PM-OPD based on small donors in Introduction.
2. Figure 1a and b should label the materials of two electrodes as well as anode and cathode. Symbols of electron and holes should be given to indicate what carriers are injected from anode and cathode under reverse bias, respectively. Figure caption should clearly state what bias is applied for Figure 1a and b. Energy levels of materials involved in the device should be provided.
3. The UV-Vis absorption spectra of ZnPc and C60 should be provided to show the absorption range of materials used in the active layer. These spectra could also help to explain feature exhibited in EQE, that is, a peak around 390 nm, a dip around 490 nm, and broad response between 500-870 nm (Figs.1d and 2a).
4. P. 4, lines 114-115. It should clearly state whether HATNA-Cl6 is also an ETL and whether it is doped or not.
5. Apparently, EQE varies with the incident light wavelength. When reporting the maximum EQE at difference reverse biases, the EQE at which wavelength should be indicated. What is the physical meaning of the ratio of EQE at different reverse biases to zero bias in Figure 1e? The calculation of specific detectivity will use the absolute value of EQE at each bias not the ratio. So, should the absolute EQE be a meaningful comparison? As author pointed out the EQE increases with the increase of applied reverse bias, why the reverse bias is capped at -10 V? Did authors try to further increase the bias?
6. P. 5, lines 139-140, please cite reference(s) in this sentence. "Recently, enhanced CT state absorption

photodetectors (CT-OPDs) have been introduced, which could benefit from high EQEs provided by photomultiplication.”

7. Dark current of the device containing 3 wt% C60 increases significantly with the applied reverse bias. Since Figure 1 displays the EQE up to -10 V reverse bias, J-V curves in Fig. 2a should also show the reverse bias to -10 V.

8. Figure 3 shows the J-V characteristics and D^* of devices based on “Photomultiplier, 3 wt%” and “pin-diode, 3 wt%”. What is the device structure of pin-diode? It is suggested to show the J-V curves under forward bias, for example, to +2 V bias. There is no rectification shown in J-V curves under illumination for a “Photomultiplier, 3 wt%” device (Fig. 2a). It is interesting to show if the “pin-diode, 3 wt%” device shows a rectification in both dark and illuminated J-V characteristics.

9. The entire paragraph on pp. 10-11, lines 274-282 is very unclear. (1) Please provide the transfer matrix optical simulations results. (2) Please clearly indicate which active layer thickness corresponds to which resonant wavelength. (3) What is the exact device structure to realize the so called “optical cavity” effect.

10. Different wavelength range of EQE spectra are shown, for example, 300-900 nm (Fig. 1d and Fig. 2b), 300-1400 nm (Fig. 4a), and 600-1200 nm (Fig. 4c). Any explanations?

11. The concept of charge collection narrowing (CNN) was first proposed in the paper of ref. 42, not ref. 33.

12. P. 13, line 347, “The intensity is controlled by a Hamamatu S1337 silicon photodiode”. The intensity is calibrated by this standard diode not controlled by this standard diode.

13. The active layers were prepared via vapor deposition, which gives an handle for controlling the distribution of C60 in the active layer relatively easier than using spin coating method, for example, a gradient composition distribution with more C60 in the active layer close to the cathode to facilitate electron trapping and band bending. Therefore, deposition conditions are critical and the details for organic molecules (ZnPc and C60) should be provided, such as how to control the composition of C60 to be 1 to 4 wt%, what are the deposition rates, what are the thickness of active layers, what is the background pressure, etc.

14. The ITO layer is quite thin. What is the conductivity of 90 nm ITO/glass?

Reviewer #3 (Remarks to the Author):

The manuscript describes a vacuum-processed organic photodetector, claimed to operate based on the photomultiplication principle, that leverages near-infrared absorption of a charge-transfer state between the two small molecular organic compounds comprising the absorption layer. The reported device performance is quite good on some of the relevant metrics, and the principle of operation potentially enhances the path to new and existing design and application possibilities.

The paper is generally well written and can be followed, albeit non-specialists may have to spend extra time parsing some of the specialized photodetector terminology. Figures are well formatted, presenting adequately the principle of operation, schematically showing the composition, and most of the key photoresponse characteristics of the device.

The concept itself appears to be a synthesis of prior work on photomultiplication detection in the solid state and organic photodetectors leveraging optical cavity-enhanced absorption, as well as some prior work showing absorption by the CT state. The specific choice of molecules is not new, although they are combined judiciously to produce a good photoresponse. Thus, the merit seems to be primarily in the practicality of the device in principle, and the path forward afforded by its processing, thinness, and low dark noise relative to alternatives.

It such light, it would be useful to better understand why the observed response is attributable to PMT action specifically, as opposed to say that of a Schottky diode. The discussion may also benefit from examining the temperature dependence of the response. Finally, the speed of operation of this device should be discussed, and whether it would place any practical constraints on its applications.

Rebuttal for:

Enhancing Sub-Bandgap External Quantum Efficiency by Photomultiplication for Narrowband Organic Near-Infrared Photodetectors

Jonas Kublitski, Axel Fischer, Shen Xing, Lukasz Baisinger, Eva Bittrich, Johannes Benduhn, Donato Spoltore, Koen Vandewal, Karl Leo

NCOMMS-20-44275, March 29th, 2021.

Dear reviewers,

We thank you for your careful peer-review of our manuscript and the very helpful criticism. In response to the reviewer's comments, we made substantial changes to the manuscript. In the document below, all the concerns and points raised are addressed and additional measurements and analyses are presented. For better readability, we visually structured our response as following:

The comments and points of the referees are copied in black color and italic style.

Our response is colored in green.

Furthermore, we cite parts from the revised manuscript or the revised supporting information to afford a quick impression of the additions and improvements to the reviewers. These citations are indicated by blue text and are shifted to the right.

We hope that all points are clearly addressed.

Sincerely,

Jonas Kublitski on behalf of all authors

Reviewer #1 (Remarks to the Author):

Kublitski et al. reported demonstrate fully vacuum-processed narrowband and broadband PM-OPDs with ZnPc:C₆₀ (100:3, wt/wt) as the active layers. The active layers have single carrier transport channels and some isolated electron traps formed by ZnPC/C₆₀/ZnPc. The photogenerated electrons will be trapped in C₆₀, which will induced the interfacial band bending for hole tunneling injection from external circuit. Devices are based on enhanced hole injection leading to an maximum external quantum efficiency (EQE) of almost 2000% at -10 V for the broadband device. I am pleased to recommend this manuscript potential publication in Nature Communication after well addressing the following issues, please note that the statement on PM in Near infrared region is wrong due to the EQE values are much less than 100%.

We thank the reviewer for the fruitful comments and for recommending our manuscript for publication. Below we address the question raised:

1. *The authors claimed that “Narrowband PM-OPDs show peak EQEs from 20 to 80% under -10 V with full width at half maximum (FWHM) from 20 to 40 nm” is wrong. If the EQEs are lower than 100%, which is not a kind of PM-OPDs, should be photodiode type organic photodetectors.*

While we understand the concerns raised regarding the definition of the photomultiplication (PM) effect, we believe that PM not necessarily has to lead to EQEs above 100%, since it also depends on the absorption characteristics of the targeted region. Especially for charge-transfer states, it is well documented that the absorption cross-section of these states is orders of magnitude lower than the main absorption region. Therefore, as a lower portion of the incoming photons is absorbed, less free charge carriers are generated, decreasing the magnitude of the EQE. A more robust characterization of the PM effect in the sub-gap absorption region can be achieved by analyzing the internal quantum efficiency (IQE), which takes into account only absorbed photons. Indeed, in Supplementary Figure 4b, we show that the calculated IQE reaches ~160% at -10 V for the PM-OPD with detection wavelength of 828 nm, even though the

EQE is 80%. Nonetheless, it is also possible to demonstrate the PM simply by further increasing the applied voltage. This is summarized in Supplementary Figure 4, where *EQEs* over 100% are demonstrated for all the devices at -15 V leading to *IQEs* of 920% for the best performing device. We discuss this point in lines 46 to 49 of the revised manuscript:

“If the transit time of injected charge carrier is lower than the lifetime of the accumulated, photo-generated charges, an *EQE* > 100% is observed. Here, we would like to stress that prior to the photomultiplication process the photon needs to be absorbed by the active layer. We therefore conclude that the minimum criteria for photomultiplication is that the internal quantum efficiency (*IQE*) is larger than unity.”

In lines 84 to 92:

“Such narrowband OPDs could significantly benefit from the increased *IQE*, if photomultiplication would take place also by direct excitation of CT states. However, it is unclear whether direct excitation of CT states can result in a photomultiplication process. Utilizing the intermolecular CT state absorption renders possible to detect NIR photons beyond 1700 nm meanwhile using rather small and sublimable organic semiconductors where the absorption profile can be easily tuned by the D-A system. On the other hand, the weak intermolecular absorption cross section^{1,2} challenges the overall performance and here the photomultiplication could improve the electrical performance by increasing the gain for every absorbed photon.”

Also in lines 296 to 303:

“While the PM effect is commonly accompanied by an *EQE* above 100%, it is the *IQE* which better defines the physical phenomenon behind this effect. In order to induce PM, free charge carriers must be firstly generated, requiring photons to be absorbed. As a means of quantifying whether absorbed photon induce enhanced injection in the sub-gap absorption region in our devices, we estimated their *IQE*, which accounts only for absorbed photons. To that end, we employ the transfer

matrix method^{3,4} (TMM) using ellipsometrically derived n, k -values to simulate the absorption in our devices and adjust the IQE to reproduce the magnitude of the measured EQE spectra, see Supplementary Figure 6 for more details.”

Supplementary Figure 6: n, k -values of ZnPc:C₆₀ (3 wt%). (a) In-plane n -values are obtained from variable-angle spectroscopic ellipsometry as described in the main text. The absorption in the CT state absorption region is very weak, hindering the modeling of k -values from spectroscopic ellipsometry. As we are mainly interested in this region, we derived k -values as described by Kaiser *et al.*¹. The method requires device parameters such as EQE spectrum, which is obtained from a pin-diode with the same structure shown in Supplementary Figure 2b, but with active layer thickness of 100 nm. Since the architecture of the pin-diode is not optimized, the EQE is measured at -10 V to ensure that all photogenerated charges are extracted. Additionally, n, k -values of all other layers in the device must be provided, which were obtained by spectroscopic ellipsometry. In (a) the simulated k -values (red solid line) are compared to the measured k -values (dashed line) in the visible range. A good agreement is achieved, indicating that the simulated k -values of ZnPc:C₆₀ (3 wt%) according to Kaiser *et al.*¹ are properly derived. (b) Measured EQE at -10 V of the pin-diode compared to the TMM simulated spectrum of the same device using the n, k -values from (a).

And in lines 319 to 325 of the main text:

“As to prove that photomultiplication also takes place in the narrowband devices, we estimate the IQE of these devices. Indeed, for the device with a detection wavelength of 828 nm, an IQE of 160% is achieved. The three other devices show IQE of around 40%, from which it is not possible to infer whether such values are a result of the PM effect or other phenomena. Therefore, to elucidate that, we increase the bias voltage to -15 V. This leads to $EQEs$ and $IQEs$ above 100% for all four devices, with peak values of $\sim 430\%$ and $\sim 920\%$, respectively, see Supplementary Figure 4.”

Supplementary Figure 4: EQE and IQE of narrowband devices. (a) EQE of narrowband devices measured at -15 V. (b) Estimated IQE at -15 V (dashed black line) and -10 V (red dashed line). At -15 V both (a) EQE and (b) IQE of all narrowband devices are higher than 100%, demonstrating that photomultiplication is achieved for these devices. In (b), dashed lines are guide to the eye.

2. The authors also claimed “narrowband photomultiplication type devices based on charge collection narrowing (CCN)³³” The reference [33] reported narrowband photomultiplication type organic photodetectors based on charge injection narrowing (CIN) concept, rather than CCN concept. The detailed description on CIN concept is given in *Adv. Opt. Mater.* 2018, 1800249. The difference between CCN and CIN should be carefully discussed for the reader better understand. The OPDs based on CCN also exhibit EQE much less than 100%, the narrowband is obtained at the price of EQE. The narrowband prosperity of OPDs based on CCN will lose under large bias. For the OPDs based on CIN, the EQE will be markedly increased along with the increase of bias, while keeping the FWHM as constant. The typical works on PM-OPDs, *Org. Electron.* 2019, 69, 354, *Nanoscale* 2019, 11, 16406 should be cited, the active layers also have single charge carrier transport and some isolated the opposite charge traps.

We thank the reviewer for pointing out this misunderstanding. The description of the effect demonstrated in ref. [33] (now 39) was improved. In lines 105 to 107 in the main text:

“These results are comparable with narrowband organic pin-photodiodes based on cavities^{5,6}, and higher than that of narrowband photomultiplication type devices based on charge injection narrowing (CIN)⁷.”

In lines 332 to 334:

“Moreover, it is superior than that of narrowband photomultiplication type devices based on CIN⁷, where, in addition, excessively thick devices are demanded, which increases the operation voltage.”

In lines 358 to 362:

“In CT-OPDs, the thicknesses required are much smaller than those used in narrowband devices based on charge collection narrowing (CCN)^{8,9} or on CIN^{7,10}. In the latter, for example, the thickness of the active layer must be much larger than the inverse of the absorption coefficient of the active layer, such that under illumination charges are generated close to the injecting contact, thereby causing the necessary band bending¹⁰.”

In lines 365 to 368:

..., where active layers of 355 nm were used for spectral response at ~830 nm, compared to 2.5 μm reported for spectral response at 650 nm when using CIN combined with photomultiplication⁷.

The requested references are cited in lines 50 to 51:

“The effect described above has been applied in organic and hybrid PDs, leading to outstanding external quantum efficiencies (EQEs) as high as 10^{5%}^{11–13}.”

And lines 41 to 43:

“...different strategies have been introduced to achieve such charge accumulation and thereby the required energy level bending through a lack of percolation path for one charge carrier type^{14–17}...”

3. *The authors should investigate the OPDs with ZnPc/ZnPc:C₆₀ (100:3, wt/wt) as the active layers, I guess that the double layers OPDs will exhibit different spectral range or a narrowband response. The injected holes will be more efficiently transported in ZnPc layer, leading to the enhanced EQE values.*

As suggested in the conclusion, we believe that this approach can indeed lead to an improved performance, besides improving optical interference problems intrinsic to optical microcavity devices. However, this ongoing topic still requires a careful study, which is currently under consideration. Nonetheless, the results shown in our manuscript already describe the novel possibility of enhancing *EQE* in the CT absorption range, combined with microcavities, which had not been achieved so far.

4. *For the NIR OPDs, the EQE values are lower than 100%, which should be photodiode type OPD, rather than PM-OPDs. The authors should carefully revise the corresponding statement. The recent review article Laser Photonics Rev. 2020, 2000262 should be cited about the development on broadband organic photodetectors, including photodiode type and photomultiplication type.*

This is similar to statement 1, which we have addressed above. The topic discussed in lines 46 to 49, 84 to 92, 296 to 303 and lines 319 to 325 of the text. Thanks for pointing out reference Laser Photonics Rev. 2020, 2000262 which is now cited in line 40:

“Photomultiplication-type organic photodetectors (PM-OPDs) are capable of amplifying small photocurrents without requiring external/additional circuit components. This can be achieved by a photo-induced enhanced injection via energy level bending caused by charge accumulation near the injecting electrode^{18,19}.”

5. *The authors should pay more attention on the reference [33] mentioned many times in your manuscript, the working mechanism is not CCN, should be CIN (Charge injection narrowing). The PM should be attributed to interfacial traps induced interfacial band bending for hole tunneling injection from external circuit.*

We thank the reviewer for pointing out this misunderstanding. The description of the effect demonstrated in ref. [33] (now 41) has been improved. The wrong citations of this reference are fixed as discussed in request 2.

We also emphasized in lines 134-134 that the *EQE* enhancement arises from injected holes from the external circuit:

“This accumulation of electrons upon illumination causes that the energy levels bend in the vicinity of the contact, enabling holes from the external circuit to tunnel through the energy barrier imposed by the HATNA-Cl₆ layer into the donor phase, where they are efficiently transported together with the photo-induced holes towards the anode.”

6. The caption of Fig. 4 “*Photomultiplication in the CT region*” is wrong, it is apparent that the *EQE* values are much less than 100%. The authors should carefully revised the related points, especially for the title of this manuscript “... *Photomultiplication in Narrowband Organic Near-Infrared Photodetectors*”.

This addresses point 1, which we discussed above as in lines 46 to 49, 84 to 92, 296 to 303 and lines 319 to 325 of the text. Due to the reasons explained above, the caption of Figure 4 is kept.

Reviewer #2 (Remarks to the Author):

The authors presented a method to enhance the external quantum efficiency (EQE) of near-infrared (NIR) photoresponse. This is realized by utilizing the charge-transfer states between organic electron donor molecule (D, ZnPc) and electron acceptor molecule (A, C₆₀) with the photodiode having a photomultiplication-type structure, that is, reducing the acceptor composition to a few percent (e.g., 3 wt% here). The work mainly divided into two parts. The first part is to demonstrate the photomultiplication effect can be achieved with the active layer having 3 wt% C₆₀ in ZnPc. A maximum EQE of almost 2000% at -10V and the optimal specific detectivity D of 2.4×10¹² Jones at*

about -2.5 V were demonstrated. The idea of on-purpose reducing acceptor composition in either polymer:fullerene or polymer:semiconductor nanoparticle active layer to achieve electron-trapping induced band bending and hole injection to realize photomultiplication has been reported before (e.g., refs. 10 and 14).

Therefore, the first part of the work is an increment of the previously published work. The second part of the work is to utilize this photomultiplication-type photodiode (mainly the active layer with 3 wt% C₆₀) but with semi-transparent Ag bottom electrode and opaque Ag top electrode to form nanocavity to realize the NIR photoresponse due to charge-transfer states. By varying the active layer thickness, the wavelength of NIR response varies so does EQE. The highest EQE was achieved ~80% at the peak position of 826 nm at -10V applied bias. Utilizing charge-transfer states of ZnPC:C₆₀ and nanocavity to realize NIR detection was reported by the same group three years ago at Nature Communications (ref. 30). The difference between this work and the previous one is mainly the composition change (from roughly ZnPC:C₆₀ of 1:1 ratio to 3 wt% C₆₀) and active layer thickness change (from < 100 nm to 300-400nm). If compare the photoresponse at the same wavelength, e.g. ~870 nm, EQE is ~20% (ref. 30 Fig. 2a), and EQE is ~20% (this work, Fig. 4C). Therefore, the second part of the work is also an increment of the previous published work by the same group. More comments are provided below.

We thank the reviewer for the critical review which helped us to improve the quality of the manuscript. Nonetheless, we would like to point out that there are fundamental differences between previous reported work⁵ and what is being proposed in this manuscript. Previously, Siegmund *et al.*⁵ reported a diode-type OPD embed in a cavity. Such devices are limited to EQEs of 100%. The devices reported here combine the photomultiplicative gain with an optical cavity, allowing for EQEs higher than 100% even in the sub-band gap region. In order to explain this effect in more detail, we added Supplementary Figure 4, where, besides showing EQE over 100% for all narrowband devices at -15 V, we also estimate the internal quantum efficiency of our devices. The PM-OPDs demonstrated here are capable of providing IQEs over 100%, which is by definition not possible in the work by Siegmund *et al.*⁵

Below we address the questions and comments raised by the reviewer:

1. *The PM-OPDs discussed in introduction are mainly polymeric donors, few are small molecule donors (or not explicitly discussed). This work used small molecule as donor. So, it would be helpful a review of the PM-OPD based on small donors in Introduction.*

Most PM-OPDs realized so far comprise polymers as active layer. In fact, the fabrication of small molecule PM-OPDs based on low acceptor content was not demonstrated so far. Although some work has been performed on vacuum-processed materials as active layer, the working principle is substantially different, using interfacial layer to cause charge carrier accumulation. Nonetheless, we extended the discussion of the related publications in lines 65 to 75:

“Photomultiplication has been extensively exploited in solution-processed organic/hybrid devices. However, despite the many advantages offered by sublimable small molecules, fewer examples were demonstrated in fully vacuum-processed devices^{20–24}. Huang *et al.* demonstrated *EQE* higher than 1000% in devices based on C₆₀. These values were attributed to the disordered structure of C₆₀ and to interfacial traps at the interface C₆₀/hole transporting layer²⁵. Similar results were achieved by interfacial blocking layers in hybrid (solution- and vacuum-processed)^{26,27} and fully vacuum-processed devices^{28,29}, which are used to avoid charge extraction, thereby causing the necessary band bending. In general, the vacuum deposition provides the possibility of depositing a vertical gradient of donor or acceptor molecules in the blend, as well as fine tuning the mixing ratio. Yet, such fine tuning extensively used in solution-processed PM-OPDs has not been investigated in vacuum-processed devices. Besides that, vacuum deposition offers the possibility of sequentially stacking of multiple layers, the well-established doping technology^{30,31}, straightforward fabrication of matrices of individual pixels, and is for commercial organic optoelectronic devices the currently preferred manufacturing technique.”

2. *Figure 1a and b should label the materials of two electrodes as well as anode and cathode. Symbols of electron and holes should be given to indicate what carriers are*

injected from anode and cathode under reverse bias, respectively. Figure caption should clearly state what bias is applied for Figure 1a and b. Energy levels of materials involved in the device should be provided.

Thank you for pointing out these deficiencies. The figure and the caption were updated to better explain the device working principle.

Figure 1 | Operation, device structure and EQE of a PM-OPD. Schematic energy diagram **a**, under dark at flat band condition and **b**, negatively biased under illumination. **c**, schematic device structure. **d**, Voltage-dependent EQE (solid lines) of the device shown in **(c)** comprising ZnPc blended with C₆₀ at 3 wt%. Each line corresponds to one symbol in **(d)**. Dashed red line shows the absorption spectrum of the same blend. Additionally, in **e**, the relative enhancement factor as a function of applied reverse bias is presented. Symbols show the ratio between EQE at 670 nm at each voltage from **(d)** normalized by the EQE at 670 nm at 0 V. The blue line is a guide to the eye. Note that no saturation is observed, indicating that EQE can be further increased. Inset shows the chemical structure of C₆₀, ZnPc and HATNA-Cl₆. The energy level values of ZnPc, C₆₀ and HATNA-Cl₆ in **(a)** are taken from the literature³²⁻³⁴.

3. The UV-Vis absorption spectra of ZnPc and C₆₀ should be provided to show the absorption range of materials used in the active layer. These spectra could also help to explain feature exhibited in EQE, that is, a peak around 390 nm, a dip around 490 nm, and broad response between 500-870 nm (Figs. 1d and 2a).

The UV-Vis absorption spectrum of the blend ZnPc:C₆₀ is now shown together with the voltage-dependent EQE in Fig. 1d.

Figure 2 | Operation, device structure and EQE of a PM-OPD. Schematic energy diagram **a**, under dark at flat band condition and **b**, negatively biased under illumination. **c**, schematic device structure. **d**, Voltage-dependent EQE (solid lines) of the device shown in **(c)** comprising ZnPc blended with C₆₀ at 3 wt%. Each line corresponds to one symbol in **(d)**. Dashed red line shows the absorption spectrum of the same blend. Additionally, in **e**, the relative enhancement factor as a function of applied reverse bias is presented. Symbols show the ratio between EQE at 670 nm at each voltage from **(d)** normalized by the EQE at 670 nm at 0 V. The blue line is a guide to the eye. Note that no saturation is observed, indicating that EQE can be further increased. Inset shows the chemical structure of C₆₀, ZnPc and HATNA-Cl₆. The energy level values of ZnPc, C₆₀ and HATNA-Cl₆ in **(a)** are taken from the literature³²⁻³⁴.

4. P. 4, lines 114-115. It should clearly state whether HATNA-Cl₆ is also an ETL and whether it is doped or not.

HATNA-Cl₆ is used intentionally undoped to achieve charge accumulation near the contact. While its low HOMO of around -7.1 eV blocks hole injection under reverse bias in the dark, its low conductivity helps the necessary electron accumulation. This is further clarified in the line 123:

“Pristine HATNA-Cl₆ is used as hole blocking layer (EBL)”

And lines 131 to 134 in the main text:

“...electrically undoped HATNA-Cl₆ layer³⁵. While n-doped HATNA-Cl₆ has been already employed as an electron transport layer, in this device, we intentionally use a pristine layer such that the electron extraction is hindered and slowed down, which helps the electron accumulation at the cathode.”

5. *Apparently, EQE varies with the incident light wavelength. When reporting the maximum EQE at difference reverse biases, the EQE at which wavelength should be indicated. What is the physical meaning of the ratio of EQE at different reverse biases to zero bias in Figure 1e? The calculation of specific detectivity will use the absolute value of EQE at each bias not the ratio. So, should the absolute EQE be a meaningful comparison? As author pointed out the EQE increases with the increase of applied reverse bias, why the reverse bias is capped at -10 V? Did authors try to further increase the bias?*

For clarity, we now specify the wavelength at which EQE is extracted, namely 670 nm, which represents the maximum EQE for all voltages applied. Indeed, in calculation of the specific detectivity D^* , only the absolute value of EQE is needed as is shown in Fig. 3b, right axis. However, Fig. 1e represents the relative increase in EQE upon bias increase, as compared to the value achieved without enhanced charge injection.

As shown in Fig. 3b, increasing the reverse applied voltage does not increase the detectivity of our device, as the dark current increases concomitantly. Therefore, in spite

of an expected increase in EQE, we capped the voltage at -10 V. We briefly explain that in line 148 to 149:

“However, as it will be discussed below, an optimum operation regime exists in the range of -2.5 V, where the highest D^* is achieved.”

6. P. 5, lines 139-140, please cite reference(s) in this sentence. “Recently, enhanced CT state absorption photodetectors (CT-OPDs) have been introduced, which could benefit from high EQEs provided by photomultiplication.”

The references are included in line 164:

“Recently, enhanced CT state absorption photodetectors (CT-OPDs) have been introduced^{5,6,36}, which could benefit from high gain for absorbed photons provided by photomultiplication. Recently, enhanced CT state absorption photodetectors (CT-OPDs) have been introduced^{5,6,36}, which could benefit from high gain for absorbed photons provided by photomultiplication.”

7. Dark current of the device containing 3 wt% C_{60} increases significantly with the applied reverse bias. Since Figure 1 displays the EQE up to -10 V reverse bias, J-V curves in Fig. 2a should also show the reverse bias to -10 V.

The JV curve for 3 wt% device, which shows the best performance, is shown in Fig. 3a and was measured until -10 V. Fig. 2 instead aims to explain the difference between different acceptor concentrations, we believe therefore that the presented data is conclusive.

8. Figure 3 shows the J-V characteristics and D^* of devices based on “Photomultiplier, 3 wt%” and “pin-diode, 3 wt%”. What is the device structure of pin-diode? It is suggested to show the J-V curves under forward bias, for example, to +2 V bias. There is no rectification shown in J-V curves under illumination for a “Photomultiplier, 3 wt%” device (Fig. 2a). It is interesting to show if the “pin-diode, 3 wt%” device shows a rectification in both dark and illuminated J-V characteristics.

Supplementary Figure 1 was included, where the structure of the pin-diode as well as the JV curves showing the rectification under forward bias are visualized.

Supplementary Figure 2: pin-diode. (a) JV curves under dark and under 100 mW cm^{-2} illumination. (b) structure of the pin-diode.

9. The entire paragraph on pp. 10-11, lines 274-282 is very unclear. (1) Please provide the transfer matrix optical simulations results. (2) Please clearly indicate which active layer thickness corresponds to which resonant wavelength. (3) What is the exact device structure to realize the so called “optical cavity” effect.

We apologize for the rather confusing discussion. We re-structured the referred paragraph from line 304 to 322. Additionally, the structure of the narrowband PM-OPD is shown in the inset of Figure 4c and the legend states the thickness for each respective resonant wavelength. The optical simulation of the EQE is shown in Supplementary Figure 3.

“In order to test whether such devices could be achieved, we embedded the best performing PM-OPD, i.e. 3 wt%, into an optical microcavity, see inset in Figure 4 for the device structure. Due to the higher Ag work function as compared to that of ITO, we inserted a 10 nm thick MeO-TPD layer to hinder hole injection in reverse bias. With aid of TMM, we simulate the optical photoresponse of a device comprising the same active layer thickness of 400 nm, which leads to a resonant peak around 880 nm. Different resonant peaks can be achieved by varying the thickness of the active layer, leading to tunable near infrared detection as shown

in Supplementary Figure 3. The JV and EQE characteristics of devices comprising thicknesses from 355 nm to 400 nm are shown in Figure 4b and Figure 4c, respectively. As predicted by the optical simulation, narrowband peaks arise in the EQE spectra. As a demonstration, we tune the response wavelength from ~ 830 nm to ~ 880 nm, which under -10 V, reaches maximum EQE of 20% to 80%, with a FWHM varying from 20 nm to 40 nm. As to prove that photomultiplication also takes place in the narrowband devices, we estimate the IQE of these devices. Indeed, for the device with a detection wavelength of 828 nm, an IQE of 160% is achieved. The three other devices show IQE of around 40%, from which it is not possible to infer whether such values are a result of the PM effect or other phenomena. Therefore, to elucidate that, we increase the bias voltage to -15 V. This leads to $EQEs$ and $IQEs$ above 100% for all four devices, with peak values of $\sim 430\%$ and $\sim 920\%$, respectively, see Supplementary Figure 4.”

Supplementary Figure 4: EQE and IQE of narrowband devices. (a) EQE of narrowband devices measured at -15 V. (b) Estimated IQE at -15 V (dashed black line) and -10 V (red dashed line). At -15 V both (a) EQE and (b) IQE of all narrowband devices are higher than 100%, demonstrating that photomultiplication is achieved for these devices. In (b), dashed lines are guide to the eye.

Figure 3 | Photomultiplication in the CT region and used in narrowband devices. a, EQE as a function of excitation wavelength for the broadband photodetector. Red dashed line shows the EQE spectrum of a conventional ZnPc:C₆₀ (50 wt%) photodiode and the black solid line the PM-OPD, both at 0 V. Solid lines show the spectra of the PM-OPD (3 wt% C₆₀) under bias as indicated in the legend. Under -5 V, the higher injection provided by the energy level bending leads to an EQE higher than that of a conventional pin-photodiode. **b**, JV characteristics under dark and under 100 mW cm⁻² of four different narrowband devices with varying resonant wavelength as indicated in the legend. **c**, EQE of cavities of the same devices shown in **b** at -10 V. As the active layer thickness increases from 355 nm to 400 nm, the resonant wavelength redshifts from around 830 nm to 880 nm. Dashed lines show the fit to a Lorentzian function, from which the FWHM is extracted. **d**, EQE measured at different voltages is fitted with polynomial function, from which D^* is predicted. An optimum operation region is found around -3.5 V, where D^* of 6×10^{11} Jones is obtained.

Supplementary Figure 3: TMM simulation of narrowband devices. Normalized simulated EQE for four different thicknesses (d) of the optical microcavity, leading to four detection wavelengths (λ_{detec}), as presented in the main text.

10. Different wavelength range of EQE spectra are shown, for example, 300-900 nm (Fig. 1d and Fig. 2b), 300-1400 nm (Fig. 4a), and 600-1200 nm (Fig. 4c). Any explanations?

The sensitivity of the setup used in the measurement of Fig. 1d and Fig. 2b is much lower than the one used for the measurements shown in Fig. 4a and Fig. 4c, so that no signal can be measured for wavelength above 900 nm. This setup also uses only a Si diode, in contrast to the more sensitive one, which also uses an InGaAs diode as reference. In Fig. 4a, the entire measurable spectra are shown in order to convince the reader that the amplification occurs similarly from 300 to 1400 nm. In Fig. 4c, we focus on the region for which the cavity effect is observed.

11. The concept of charge collection narrowing (CNN) was first proposed in the paper of ref. 42, not ref. 33.

Thank you for pointing out this error. The reference has been fixed in line 358.

“In CT-OPDs, the thicknesses required are much smaller than those used in narrowband devices based on charge collection narrowing (CCN)^{8,9} or on CIN^{7,10}.”

12. P. 13, line 347, “The intensity is controlled by a Hamamatsu S1337 silicon photodiode”. The intensity is calibrated by this standard diode not controlled by this standard diode.

The corresponding sentence has been rewritten in line 405:

“The intensity is calibrated by a Hamamatsu S1337 silicon photodiode.”

13. The active layers were prepared via vapor deposition, which gives a handle for controlling the distribution of C₆₀ in the active layer relatively easier than using spin coating method, for example, a gradient composition distribution with more C₆₀ in the active layer close to the cathode to facilitate electron trapping and band bending. Therefore, deposition conditions are critical and the details for organic molecules (ZnPc and C₆₀) should be provided, such as how to control the composition of C₆₀ to be 1 to 4 wt%, what are the deposition rates, what are the thickness of active layers, what is the background pressure, etc.

We briefly comment this aspect in lines 388 to 390 and included Supplementary Table 2, where the deposition conditions are shown.

“Additionally, the method presented here allows placing the active layer in different positions within the device or using gradients of D-A mixing ratio, thereby enhancing injection and diminishing the effect of optical overtones, a critical problem in CT-OPDs.”

Supplementary Table 2: Process parameters for the vacuum deposition of PM-OPDs.

Materials	Broadband				Narrowband			
	Concentration [wt%]	Thickness [nm]	Rate [\AA s^{-1}]	Vacuum chamber pressure [mbar]	Concentration [wt%]	Thickness [nm]	Rate [\AA s^{-1}]	Vacuum chamber pressure [mbar]
MoO ₃	-	-	-	-	100	3	0.2	1.5×10^{-7}
Ag	-	-	-	-	100	25	0.6	2.1×10^{-6}
MeO-TPD	-	-	-	-	100	10	0.5	4.0×10^{-7}
ZnPc	97	388	0.5	2.4×10^{-7}	97	variable	0.5	$\approx 10^{-7}$
C ₆₀	3	12	0.015	2.4×10^{-7}	3	variable	0.015	$\approx 10^{-7}$
HATNA-Cl ₆	100	10	0.5	2.7×10^{-7}	100	10	0.4	3.2×10^{-7}
Al	100	100	0.5	2.2×10^{-6}	-	-	-	-
Ag	-	-	-	-	100	100	1.0	1.8×10^{-6}

14. The ITO layer is quite thin. What is the conductivity of 90 nm ITO/glass?

The sheet resistance of ITO is $32 \Omega \square^{-1}$. This information is now provided in line 394:

“...pre-structured ITO contact ($32 \Omega \square^{-1}$, Thin Film Devices) at ultrahigh...”

Reviewer #3 (Remarks to the Author):

The manuscript describes a vacuum-processed organic photodetector, claimed to operate based on the photomultiplication principle that leverages near-infrared absorption of a charge-transfer state between the two small molecular organic compounds comprising the absorption layer. The reported device performance is quite good on some of the relevant metrics, and the principle of operation potentially enhances the path to new and existing design and application possibilities.

The paper is generally well written and can be followed, albeit non-specialists may have to spend extra time parsing some of the specialized photodetector terminology. Figures are well formatted, presenting adequately the principle of operation, schematically showing the composition, and most of the key photoresponse characteristics of the device.

The concept itself appears to be a synthesis of prior work on photomultiplication detection in the solid state and organic photodetectors leveraging optical cavity-enhanced absorption, as well as some prior work showing absorption by the CT state. The specific choice of molecules is not new, although they are combined judiciously to produce a good photoresponse. Thus, the merit seems to be primarily in the practicality of the device in principle, and the path forward afforded by its processing, thinness, and low dark noise relative to alternatives.

It such light, it would be useful to better understand why the observed response is attributable to PMT action specifically, as opposed to say that of a Schottky diode. The discussion may also benefit from examining the temperature dependence of the response. Finally, the speed of operation of this device should be discussed, and whether it would place any practical constraints on its applications.

The authors thank the reviewer for the positive feedback provided. Below, we discuss the points raised by the reviewer:

The term photomultiplication in organic devices is commonly attributed to devices that present an increased injection under applied bias as a result of a charge accumulation close to the injecting electrodes. In our devices in dark conditions, the barrier imposed for holes under reverse bias can be interpreted as a Schottky barrier, where the activation energy is related to the barrier height. However, under illumination, the injection process is dominated by the bending of the energy level, allowing holes to efficiently tunnel through the barrier. This process is characterized by an increased *EQE*, i.e., a photo-gain, which is not expected for a Schottky diode. Therefore, in such condition, the interpretation of a Schottky diode no longer describes the working mechanism of these devices. This can be seen from the activation energy E_a under

illumination as compared to E_a under dark conditions. While in the dark E_a decreases with the reverse applied voltage, as a result of a decreased barrier height, under illumination very weak voltage dependence is observed. In the latter, the band bending allows charges to tunnel the barrier such that the barrier does not play a role in the injection.

Figure R1: Activation energy at increasing reverse bias in dark conditions and under illumination.

In Supplementary Figure 5, we included transient current measurements to analyze the time response of our devices. The rise and fall constants are summarized in Supplementary Table 1 and a discussion of the temporal response is included in the main text in lines 342 to 351.

“Another important figure-of-merit of photodetectors is the response speed. In PM-OPDs, the temporal response is believed to be limited by the trapping/detrapping dynamics^{37,38}, while other processes such as charge carrier transit time should be much shorter. In order to investigate the response speed of our devices, transient current measurements are performed. The rise time (from 10% to 90% of the device saturated signal) and fall time (from 90% to 10% of the device off signal) are summarized in Supplementary Table 1. The rise time of both broad- and narrowband devices ranges from 20 μs to 600 μs , corresponding to -3 dB cut-off frequencies of $\sim 19.5 \text{ kHz}$ to $\sim 0.4 \text{ kHz}$. These values are

comparable to the best performing PM-OPDs reported so far¹⁸ and are suitable for health monitoring and video-frame-rate imaging applications.”

Supplementary Figure 5: Transient photocurrent of (a) broad- and (b) narrowband PM-OPDs.

For all measurements, 100 Hz pulse signal was used to probe the white LED, except for the broadband device at -10 V, where 50 Hz was used due to the long decay time. Switching-on and -off time constants determined as the time the device response takes to rise from 10% to 90% (on) and to fall from 90% to 10% (off) of its maximum value. The time constants are summarized in Supplementary Table 1.

Supplementary Table 1: Speed of broad- and narrowband PM-OPDs. The transient photocurrent measurements are shown in Supplementary Figures 5a and 5b, respectively. The switching-on and -off time constants of PM-OPDs are determined as the time the device takes for the response signal to rise from 10% to 90% and to fall from 90% to 10% of its maximum value, respectively. For all measurements, 100 Hz pulse signal was used to probe the white LED, except for the broadband device at -10 V, where 50 Hz was used due to the long decay time. The duration of the light pulse was long enough to reach the steady-state (5 ms), the time at which the curves are normalized. The known relation $f_{-3\text{ dB}} \simeq 0.35t_{\text{on}}^{-1}$ is used to calculate the cut-off frequencies².

Applied bias [V]	Broadband			Narrowband (843 nm)		
	t_{on} [μs]	$f_{-3\text{ dB}} \simeq 0.35t_{\text{on}}^{-1}$ [kHz]	t_{off} [μs]	t_{on} [μs]	$f_{-3\text{ dB}} \simeq 0.35t_{\text{on}}^{-1}$ [kHz]	t_{off} [μs]
-2.5	63	5.55	280	135	2.59	326
-5.0	541	0.65	539	941	0.37	613
-10.0	556	0.63	597	18	19.44	667

References from the rebuttal letter

1. Goris, L. *et al.* Absorption phenomena in organic thin films for solar cell applications investigated by photothermal deflection spectroscopy. *J. Mater. Sci.* **40**, 1413–1418 (2005).
2. Vandewal, K. *et al.* The relation between open-circuit voltage and the onset of photocurrent generation by charge-transfer absorption in polymer: Fullerene bulk heterojunction solar cells. *Adv. Funct. Mater.* **18**, 2064–2070 (2008).
3. Pettersson, L. A. A., Roman, L. S. & Inganäs, O. Modeling photocurrent action spectra of photovoltaic devices based on organic thin films. *J. Appl. Phys.* **86**, 487–496 (1999).
4. Kaiser, C., Zeiske, S., Meredith, P. & Armin, A. Determining Ultralow Absorption Coefficients of Organic Semiconductors from the Sub-Bandgap Photovoltaic External Quantum Efficiency. *Adv. Opt. Mater.* **8**, 1901542 (2020).
5. Siegmund, B. *et al.* Organic narrowband near-infrared photodetectors based on intermolecular charge-transfer absorption. *Nat. Commun.* **8**, 15421 (2017).
6. Kaiser, C. *et al.* Manipulating the Charge Transfer Absorption for Narrowband Light Detection in the Near-Infrared. *Chem. Mater.* **31**, 9325–9330 (2019).
7. Wang, W. *et al.* Highly Narrowband Photomultiplication Type Organic Photodetectors. *Nano Lett.* **17**, 1995–2002 (2017).
8. Armin, A., Jansen-van Vuuren, R. D., Kopidakis, N., Burn, P. L. & Meredith, P. Narrowband light detection via internal quantum efficiency manipulation of organic photodiodes. *Nat. Commun.* **6**, 6343 (2015).
9. Yazmaciyan, A., Meredith, P. & Armin, A. Cavity Enhanced Organic Photodiodes with Charge Collection Narrowing. *Adv. Opt. Mater.* **7**, 1–8 (2019).
10. Wang, W. *et al.* Organic Photodetectors with Gain and Broadband/Narrowband Response under Top/Bottom Illumination Conditions. *Adv. Opt. Mater.* **6**, 1800249 (2018).
11. Zhou, X. *et al.* Ultrahigh Gain Polymer Photodetectors with Spectral Response from UV to Near-Infrared Using ZnO Nanoparticles as Anode Interfacial Layer. *Adv. Funct. Mater.* **26**, 6619–6626 (2016).
12. Jang, M. S., Yoon, S., Sim, K. M., Cho, J. & Chung, D. S. Spatial Confinement of the Optical Sensitizer to Realize a Thin Film Organic Photodetector with High Detectivity and Thermal Stability. *J. Phys. Chem. Lett.* **9**, 8–12 (2018).
13. Li, X., Li, X., Wang, S. & Xiao, Y. A trap-assisted ultrasensitive near-infrared organic photomultiple photodetector based on Y-type titanylphthalocyanine nanoparticles. *J. Mater. Chem. C* **4**, 5584–5592 (2016).
14. Li, L. *et al.* Achieving EQE of 16,700% in P3HT:PC71BM based photodetectors by trap-assisted photomultiplication. *Sci. Rep.* **5**, 9181 (2015).
15. Wang, W. *et al.* Highly sensitive polymer photodetectors with a broad spectral response range from UV light to the near infrared region. *J. Mater. Chem. C* **3**, 7386–7393 (2015).
16. Zhao, Z., Wang, J., Miao, J. & Zhang, F. Photomultiplication type organic

- photodetectors with tunable spectral response range. *Org. Electron.* **69**, 354–360 (2019).
17. Miao, J., Du, M., Fang, Y. & Zhang, F. Acceptor-free photomultiplication-type organic photodetectors. *Nanoscale* **11**, 16406–16413 (2019).
 18. Miao, J. & Zhang, F. Recent Progress on Photomultiplication Type Organic Photodetectors. *Laser Photon. Rev.* **13**, 1800204 (2019).
 19. Zhao, Z., Xu, C., Niu, L., Zhang, X. & Zhang, F. Recent Progress on Broadband Organic Photodetectors and their Applications. *Laser Photon. Rev.* **14**, 2000262 (2020).
 20. Matsunobu, G., Oishi, Y., Yokoyama, M. & Hiramoto, M. High-speed multiplication-type photodetecting device using organic codeposited films. *Appl. Phys. Lett.* **81**, 1321–1322 (2002).
 21. Hammond, W. T. & Xue, J. Organic heterojunction photodiodes exhibiting low voltage, imaging-speed photocurrent gain. *Appl. Phys. Lett.* **97**, 73302 (2010).
 22. Luo, X. *et al.* Insight into trap state dynamics for exploiting current multiplication in organic photodetectors. *Phys. status solidi – Rapid Res. Lett.* **10**, 485–492 (2016).
 23. Yang, D. *et al.* Deep ultraviolet-to-NIR broad spectral response organic photodetectors with large gain. *J. Mater. Chem. C* **4**, 2160–2164 (2016).
 24. Reynaert, J., Arkhipov, V. I., Heremans, P. & Poortmans, J. Photomultiplication in Disordered Unipolar Organic Materials. *Adv. Funct. Mater.* **16**, 784–790 (2006).
 25. Huang, J. & Yang, Y. Origin of Photomultiplication in C₆₀ Based Devices. *Appl. Phys. Lett.* **91**, 203505 (2007).
 26. Fang, Y., Guo, F., Xiao, Z. & Huang, J. Large Gain, Low Noise Nanocomposite Ultraviolet Photodetectors with a Linear Dynamic Range of 120 dB. *Adv. Opt. Mater.* **2**, 348–353 (2014).
 27. Guo, F., Xiao, Z. & Huang, J. Fullerene Photodetectors with a Linear Dynamic Range of 90 dB Enabled by a Cross-Linkable Buffer Layer. *Adv. Opt. Mater.* **1**, 289–294 (2013).
 28. Guo, D., Yang, D., Zhao, J., Vadim, A. & Ma, D. Role of interfaces in controlling charge accumulation and injection in the photodetection performance of photomultiplication-type organic photodetectors. *J. Mater. Chem. C* **8**, 9024–9031 (2020).
 29. Guo, D. *et al.* Structure design and performance of photomultiplication-type organic photodetectors based on an aggregation-induced emission material. *Nanoscale* **12**, 2648–2656 (2020).
 30. Lüssem, B., Riede, M. & Leo, K. Doping of organic semiconductors. *Phys. status solidi* **210**, 9–43 (2013).
 31. Tietze, M. L. *et al.* Elementary steps in electrical doping of organic semiconductors. *Nat. Commun.* **9**, 1–8 (2018).
 32. Barlow, S. *et al.* Synthesis, ionisation potentials and electron affinities of hexaazatrinaphthylene derivatives. *Chem. Eur. J.* **13**, 3537–3547 (2007).
 33. Zhao, W. & Kahn, A. Charge transfer at n-doped organic-organic heterojunctions. *J. Appl. Phys.* **105**, 123711 (2009).
 34. Schwarze, M. *et al.* Band structure engineering in organic semiconductors. *Science* **352**, 1446–1449 (2016).
 35. Falkenberg, C., Leo, K. & Riede, M. K. Improved photocurrent by using n-doped

- 2,3,8,9,14,15-hexachloro-5,6,11,12,17,18-hexaazatrinaphthylene as optical spacer layer in p-i-n type organic solar cells. *J. Appl. Phys.* **110**, 124509 (2011).
36. Tang, Z. *et al.* Polymer:Fullerene Bimolecular Crystals for Near-Infrared Spectroscopic Photodetectors. *Adv. Mater.* **29**, 1702184 (2017).
37. Daanoune, M., Clerc, R., Flament, B. & Hirsch, L. Physics of trap assisted photomultiplication in vertical organic photoresistors. *J. Appl. Phys.* **127**, 055502 (2020).
38. Guo, F. *et al.* A nanocomposite ultraviolet photodetector based on interfacial trap-controlled charge injection. *Nat. Nanotechnol.* **7**, 798–802 (2012).

REVIEWERS' COMMENTS

Reviewer #1 (Remarks to the Author):

I am pleased to recommend this revised version acceptable for publication in Nature Communication.

Reviewer #2 (Remarks to the Author):

The authors have conducted careful revision and addressed the questions/comments raised by reviewers. The authors also stated the novelty and the difference between this work and the previous work from the group. It is recommended for publication before making the correction as described below.

Figure 1d, it is suggested to use the right y-axis for "Absorption" because Absorption is not in the same order of percentage (if authors not choosing to use (abu) for the unit of Absorbance) as EQE.

Rebuttal for:

Enhancing Sub-Bandgap External Quantum Efficiency by Photomultiplication for Narrowband Organic Near-Infrared Photodetectors

Jonas Kublitski, Axel Fischer, Shen Xing, Lukasz Baisinger, Eva Bittrich, Donato Spoltore, Johannes Benduhn, Koen Vandewal, Karl Leo

NCOMMS-20-44275A-Z, June 14th, 2021.

Dear reviewers,

We thank you for your careful peer-review of our manuscript and the very helpful criticism. In the document below, all the concerns and points raised are. For better readability, we visually structured our response as following:

The comments and points of the referees are copied in black color and italic style.

Our response is colored in green.

We hope that all points are clearly addressed.

Sincerely,

Jonas Kublitski on behalf of all authors

Reviewer #1 (Remarks to the Author):

The authors have conducted careful revision and addressed the questions/comments raised by reviewers. The authors also stated the novelty and the difference between this work and the previous work from the group. It is recommended for publication before making the correction as described below.

We thank the reviewer for recommending our manuscript for publication.

Reviewer #2 (Remarks to the Author):

Figure 1d, it is suggested to use the right y-axis for "Absorption" because Absorption is not in the same order of percentage (if authors not choosing to use (abu) for the unit of Absorbance) as EQE.

We thank the reviewer for the suggestion. The absorption in Figure 1d is now shown in the right axis with the unit cm^{-1} .